# OpenReview forum: "The Energy Loss Phenomenon in RLHF: A New Perspective on Mitigating Reward Hacking"
_ICML.cc/2025/Conference — ICML 2025 poster_

### Official Review · Reviewer_G1Re · 2025-03-13

**Overall Recommendation:** 3

**Summary:**

This paper introduces the energy loss phenomenon in RLHF, where the L1 norm difference between input and output in the final layer of LLM increases during fine-tuning, leading to reward hacking. To mitigate this, the authors propose EPPO (Energy loss-aware PPO), which penalizes energy loss during RL optimization. Experiments across various LLMs on summarization and instruction-following tasks show that EPPO improves response quality and reduces reward hacking, compared to conventional online alignment methods like PPO with KL or response length penalties.

**Claims And Evidence:**

The proposed method is empirically well-validated. However, it is unclear whether the theoretical analysis properly supports the proposed regularization term, the energy loss-based penalty, in the reward function. While the authors define an energy loss as the reduction in the L1 norm of the embedding vector after passing through the final layer and claim that its excessive increase leads to reward hacking, the proposed regularization term does not directly penalize the proposed energy loss. Instead, it ensures that the norm-size variation of the final layer's input and output vectors remains similar between the SFT and fine-tuned models. This raises concerns about the consistency between the theoretical analysis (and the paper's overall story) and the actual role of the regularization term, making the motivation and justification for the method unconvincing. Additionally, the validity of the theoretical analysis itself is questionable, as described below in Theoretical Claims.

**Essential References Not Discussed:**

To the best of my knowledge, no essential references are missing from the paper.

**Experimental Designs Or Analyses:**

The experimental design is generally sound. However, one issue stands out in Section 6.2 (Figure 7), which aims to validate the theoretical analysis. For this result to properly support the theory, the plot should not only include the energy loss $\Delta E$ but also its mean value $-\alpha$ (or $\Delta E + \alpha$), as both terms contribute to the upper bound of the mutual information in Theorem 3. Without this, it is unclear whether the empirical results truly align with the theoretical predictions.

**Methods And Evaluation Criteria:**

The proposed methods and evaluation criteria are reasonable. The experiments use diverse LLMs and well-established tasks such as summarization and instruction-following.

**Other Comments Or Suggestions:**

* The justification for calling the L1 norm difference between the final layer's input and output the "energy loss phenomenon" is unclear, as the authors seem not to establish a strong connection between this metric and physical energy concepts. The term activation decay or norm shrinkage might be more appropriate.

* Table 1 does not appear to be referenced in the main text.

**Other Strengths And Weaknesses:**

None

**Questions For Authors:**

* Instead of using the difference in L1 norm ($\Delta E$), a more straightforward approach could be to focus only on the L1 norm of the output. Have the authors considered comparing or discussing such an approach?
* In Theorem 3, is introducing the variational distribution $q$ necessary? Since $H^{out}$ is deterministically determined by $H^{in}$, it seems possible to derive a similar result more simply as: $I(H^{in}; H^{out}) = \mathcal{H}(H^{out}) \leq H(q)$, where q is a normal distribution whose standard deviation is at least that of $H^{out}$.

**Relation To Broader Scientific Literature:**

This paper studies reward hacking in RLHF, a crucial topic in LLM alignment. It particularly relates to prior works on online alignment methods, such as KL-regularized PPO. It explores an alternative approach by introducing a new regularization to constrain the norm-size variation between the SFT and fine-tuned models.

**Theoretical Claims:**

I have reviewed the theoretical analysis, including the proofs, and found several concerns about their validity and significance.
* Corollary 4: It is unclear what meaningful insight this result provides. As the authors briefly mention, when the energy loss $\Delta E$ is greater or smaller than the average ($-\alpha$), the correlation with the upper bound of the mutual information $I(X;Y)$ appears to be reversed. This raises doubts about the practical relevance of this corollary. Additionally, since \alpha itself may change as the LLM parameters are updated, it is unclear whether Corollary 4 remains valid throughout fine-tuning. This result does not seem trivially true, so a formal proof is necessary.
* Theorem 3: While I checked the proof, whether the upper bound is meaningful or sufficiently tight is unclear. For instance, a potentially tighter bound could be obtained by considering $||H^{out}|| - \mathbb{E}[||H^{out}||]$ instead of incorporating $||H^{in}||$. Moreover, it appears that the expectation term in the upper bound might cancel out with $\sigma$.

(Minor)
* The parameter σ in Theorem 3 is not explained well. Is it the standard deviation of the energy loss? Additional clarification will be beneficial regarding its interpretation and role in the bound.
* Theorem 3 states "any layer $l$," but if the model has residual connections, the Markov chain property used in the proof may not hold, making inequality (11) invalid. Does "any layer" correspond to any transformer block? For this theorem to be correct, conditions on the network architecture should be explicitly stated.
* Corollary 5 appears to be correct, but its proof needs revision. Instead of stating "Y is uniquely determined by X" in line 709, it should be "$h^{out}$ is uniquely determined by $h^{in}$." Corresponding adjustments in the surroundings are also necessary.

---

> ### Author Rebuttal · Authors · 2025-04-01
>
> Thank you for your comments. We would like to highlight that our **main contribution** is the **empirical observation of the energy loss phenomenon** and **the corresponding RL regularization design**, as acknowledged by all other reviewers (wKXn, mgdS, and mmxN). **Theoretical analysis is included as an exploratory explanation**, with its limitations explicitly discussed in the manuscript (Lines 198-206, Page 4). We acknowledge that the current analysis only applies under certain conditions, and a **rigorous explanation is beyond the scope of this paper** but remains a promising direction for future work.
>
> ---
>
> > **Q1:** Why does the regularization term penalize energy loss variation relative to the SFT model, rather than the loss itself?
>
> **RQ1:** Our design **treats the SFT model's energy loss as a state-dependent baseline**—a common RL technique—**to reduce variance from the regularization term**. By penalizing deviations from this baseline, our approach **effectively suppressing excessive increases**, **aligning well with our theoretical analysis**. The effectiveness of this design is validated empirically in Figure [R10](https://ibb.co/N6Q9TJ8Z).
>
> > **Q2:** Practical Relevance of Corollary 4.
>
> **RQ2:**   **We would like to clarify that our main contribution lies in the empirical findings and the corresponding RL regularization.** Corollary 4 **serves as an exploratory explanation** **for our findings** from the perspective of contextual mutual information. While it may not generalize to all settings, it **offers a plausible interpretation under specific conditions**.
>
> Corollary 4 shows that **if the energy loss increase stays below a threshold**, **the upper bound of contextual mutual information is reduced**, suppressing response-context relevance, offering a theoretical explanation for the emergence of reward hacking in such scenarios. **However, when increased energy loss exceeds the threshold**, **the upper bound may rise**, **but this does not necessarily improve contextual relevance**. Why excessively increased energy loss continues to be associated with reward hacking in such scenarios remains an open question.
>
> Thus, **Corollary 4 offers a theoretical insight into reward hacking as energy loss increases**, though it is limited to a specific regime. **A more rigorous theoretical explanation is beyond the scope** of this paper but is a promising area for future research.
>
> > **Q3:** Dynamic $\alpha$ in corollary 4 during fine-tuning.
>
> **RQ3:** We would like to clarify that Corollary 4 is **not a strict theoretical guarantee for the proposed RL regularization during fine-tuning**, but rather a lens for interpreting empirical observations.  While it is true that $\alpha$ may shift as training progresses, the underlying relationship it describes still holds.
>
> > **Q4:** Questions about the upper bound of mutual information derived in Theorem 3.
>
> **RQ4:** We would like to clarify that the goal of Theorem 3 is to provide a theoretical perspective that **helps explain our empirical observations, rather than deriving the tightest possible upper bound on contextual mutual information.**
>
> While your suggested formulations, $||H^{\text{out}}||_1 - \mathbb{E}[||H^{\text{out}}||_1]$ and $H(q)$, may indeed yield a tighter upper bound, they are **significantly more difficult to estimate and optimize, particularly during dynamic fine-tuning**. In contrast, **our use of $||H^{\text{out}}||_1 - ||H^{\text{in}}||_1$ provides a more tractable and empirically estimable surrogate that still captures the core trend central to our analysis.**
>
> > **Q5:** More rigorous statement in Theorem 3’s condition and Corollary 5’s proof.  & Suggestion for using ”activation decay” & Definition of $\sigma$.
>
> **RQ5:** Thank you for your valuable suggestion. We will revise the corresponding statements and definitions for greater clarity in the revised version.
>
> > **Q6:** Question about Figure 7.
>
> **RQ6:**   We would like to clarify that Figure 7 is **not intended to validate the theoretical analysis**. Rather, it is inspired by the theoretical insights and aims to empirically investigate further, from the perspective of contextual relevance, why excessive increases in energy loss are accompanied by reward hacking.
>
> > **Q7:** Comparison with directly penalizing the L1 norm of the output.
>
> **RQ7:**  Thank you for your comments. Following your suggestion, we **conducted additional experiments on Llama3-8b to compare our approach with directly penalizing the output L1 norm**. The results, shown in Figure [R11](https://ibb.co/RTScfjPx), **demonstrate the advantage of our approach**.
>
> We hypothesize that by penalizing the L1 norm difference between the input and output of the final layer, our approach **constrains only the final layer**, thereby **preserving optimization flexibility**. In contrast, directly penalizing the output norm **inevitably constrains the preceding layers, limiting the model’s optimization capacity**.

---

> > ### Comment · Reviewer_G1Re · 2025-04-05
> >
> > Thank you for the detailed response. While I’m not seeking full theoretical analysis, I still find a disconnect between the mathematical motivation and the actual design of the proposed method.

---

> > > ### Author Response · Authors · 2025-04-05
> > >
> > > Thank you for your reply. We understand your concerns. However, we would like to clarify that **our method’s design does not build on the theoretical analysis**. Rather, the theoretical analysis is provided solely as an exploratory explanation of our empirical observations, while our method's design is entirely based on these empirical observations. As we clearly indicated in Section 1 of introduction (Line 78–79: “To address this phenomenon”) and in Section 4 of our method design (Line 216: “Building on this observation,”), these empirical observations form the foundation of our proposed method.
> > >
> > > Moreover, to further address your concern, **we have refined our theoretical analysis, which can now explain our empirical observations across all scenarios**. The updated theorem and its corresponding proof are provided in Figure [R12](https://ibb.co/zhRq2Gdn), **with the specific revisions highlighted in blue**.
> > >
> > > In the revised theorem, we theoretically demonstrate that **the L1 norm of the final output hidden state from the LLM provides an upper bound on the contextual relevance of its responses**. Therefore, during the RL process, as the energy loss in the final layer of the LLM significantly increases, the L1 norm of its output hidden state correspondingly decreases. This reduction tends to compress the mutual information between the context and the response, which may potentially lead to hacking behavior.
> > >
> > >  If any additional points require clarification or further adjustments, please do not hesitate to let us know.

---

### Official Review · Reviewer_wKXn · 2025-03-14

**Overall Recommendation:** 5

**Summary:**

This paper identifies the Energy Loss Phenomenon in RLHF, where increasing energy loss in the final layer of LLMs signals reward hacking, and provides a theoretical framework showing how this increase lowers response-context relevance, a key factor in reward hacking. To address this issue, The authors propose EPPO (Energy loss-aware PPO), a novel algorithm that penalizes the increase in energy loss during reward calculation to mitigate reward hacking. Extensive experiments show that EPPO effectively mitigates reward hacking, improves RLHF performance, and outperforms existing algorithms. The authors also demonstrate that EPPO can be viewed as an entropy-regularized RL algorithm, offering deeper insights into its effectiveness.

**Claims And Evidence:**

Yes, the claims are supported by clear and convincing evidence.

**Essential References Not Discussed:**

No, all related works have been discussed in the paper.

**Experimental Designs Or Analyses:**

Yes, I have checked the soundness and validity of the experiments in this paper.

**Methods And Evaluation Criteria:**

Yes, the proposed methods and evaluation criteria make sense for the problem at hand.

**Other Comments Or Suggestions:**

There is a typo in the caption of Figure 14 in the appendix:
AlpacaFarm, Anthropic-Helpful, Anthropic-Harmless, Reddit TL;DR datasets -> AlpacaFarm, Anthropic-Helpful, Anthropic-Harmless, and Reddit TL;DR datasets.

**Other Strengths And Weaknesses:**

Strengths:

1.	The paper is well-organized and is easy to understand.
2.	The identification of the energy loss phenomenon in LLMs is a novel contribution, offering fresh insights into the problem of reward hacking in RLHF.
3.	The theoretical explanations for the energy loss phenomenon and the proposed method are insightful.
4.	The experimental results for the energy loss phenomenon and the proposed EPPO are thorough and compelling, highlighting their potential for practical applicability in real-world scenarios.

Weakness:

1.	Why does the comparison method PPO with length penalty still suffer from reward hacking in the later stages of reinforcement learning, as shown in Figure 5? This is confusing to me.
2.	What specific advantages does the proposed EPPO algorithm have over the ensemble-based methods in [1], which are also designed to mitigate reward hacking?
3.	The paper fails to discuss the potential computational overhead (time cost) introduced by the energy loss calculation during RL process.
4.	Is there a simple and intuitive reason why the L1 norm is used in this work instead of the L2 norm?

[1] Reward model ensembles help mitigate overoptimization. ICLR 2024

**Questions For Authors:**

See Weakness.

**Relation To Broader Scientific Literature:**

This paper advances RLHF research by identifying energy loss in LLMs as a previously under-explored factor in reward hacking and introducing EPPO to effectively mitigate it.

**Theoretical Claims:**

Yes, I have checked the correctness of Theorem 3 and Corollary 5.

---

> ### Author Rebuttal · Authors · 2025-04-01
>
> We appreciate your positive feedback on the clarity, novelty, theoretical insights, and compelling experimental results of our paper. We will address each of your comments and concerns below and also in our revised manuscript.
>
> ---
>
> > **Q1:** Why does the comparison method PPO with length penalty still suffer from reward hacking？
> >
>
> **RQ1:** Thank you for your valuable feedback. As discussed in Lines 367-369 on Page 7 of the manuscript, **PPO with length penalty can only capture hacking patterns characterized by longer response lengths**, such as excessive redundancy. However, **it is not effective in addressing hacking patterns that are unrelated to response length**, such as excessive caution. Therefore, **PPO with length penalty still suffers from reward hacking to some extent**. The related case study can be found in Appendix L of the manuscript.
>
> > **Q2:** What specific advantages does the proposed EPPO algorithm have over the ensemble-based methods?
> >
>
> **RQ2:** Thank you for your insightful question. As discussed in Lines 292-296 on Page 6 of the manuscript, although the ensemble-based RM method can significantly enhance the robustness of reward modeling, **it may still be susceptible to spurious features in reward modeling and distribution shifts during the RL stage, potentially leading to reward hacking**. In contrast, our proposed EPPO algorithm **directly focuses on the neuron behavior within the LLM that is related to reward hacking**, **enabling more effective mitigation**. Furthermore, **our EPPO is more efficient compared to ensemble-based RM methods**, as the latter often requires loading multiple reward models during the RL process, which significantly increases resource demands, especially for large-scale LLMs.
>
> > **Q3:** Computational overhead analysis.
> >
>
> **RQ3:** Thank you for your valuable suggestion. Following your advice, we report the training times for both PPO and EPPO on two tasks, using a single node equipped with 8 A100-SXM80GB GPUs in the table below. As shown, **while our EPPO does incur a slight increase in training time, it significantly improves training performance**, as demonstrated in our paper.
>
> | Tasks | PPO | EPPO |
> | --- | --- | --- |
> |  General Dialogue | 7h 26min | 7h 45min |
> | Summarization | 3h 53min | 4h 06min |
>
> > **Q4:** Intuitive reason why the L1 norm is used in this work instead of the L2 norm.
> >
>
> **RQ4:** Thank you for your insightful question. The primary consideration for using the L1 norm instead of the L2 norm is that the **L1 norm is more robust to outliers**. This robustness to outliers allows the L1 norm to more reliably identify relevant features while minimizing the impact of noise, making it a more appropriate choice for analyzing the hidden state representations in LLMs.

---

> > ### Comment · Reviewer_wKXn · 2025-04-08
> >
> > Thanks for the authors' responses which address all of my concerns.

---

> > > ### Author Response · Authors · 2025-04-09
> > >
> > > Dear Reviewer wKXn,
> > >
> > > We are very glad that our response addressed your concerns. We truly appreciate your valuable feedback and the positive support for our work.
> > >
> > > Best regards,
> > >
> > > The authors of Paper 3248

---

### Official Review · Reviewer_mgdS · 2025-03-14

**Overall Recommendation:** 4

**Summary:**

This paper observes that the energy loss in the last layer of LLMs tends to get larger when using RL to train. Other than this finding, the authors also give theoretical analysis, which shows that under mild conditions, the increased energy loss reduces the upper bound of contextual relevance in LLMs. This will more easily make LLMs overfit to reward model-favored patterns in RL, which is harmful to the performance. To solve this problem, the authors provide an energy loss-aware PPO algorithm that aims to give the energy loss a penalty when it goes larger. Experiments show that the proposed approach receives good results for multiple different LLMs and it can also improve the performance of the RLHF.

**Claims And Evidence:**

The authors have well supported the claims made.

**Essential References Not Discussed:**

This paper talks about the energy loss problem of LLMs. Relevant references have been cited.

**Experimental Designs Or Analyses:**

- It is good to see the results on Llama3-8B, Mistral-7B, and DeepSeek-7B. Compared to OpenAI GPT-1o and DeepSeek-V3, these models are not large enough to support the claims made by the authors. The question is: Have the authors attempted to observe whether similar phenomenon happens to larger-sized models?

- Another question I am interested in is that: What would the model performance change when the chain of thoughts strategy is used. As the energy loss gets larger, it seems that the model performance with deep thinking would reduce accordingly. I am just curious about this.

**Methods And Evaluation Criteria:**

- Overall speaking, the observation of this paper is interesting. According to the analysis and experiments from the paper, it is really an interesting problem, which is faced by multiple LLMs.

- Though the proposed method is relatively simple, it can well address the issue posed by the authors. According to the experimental results, the proposed approach indeed works for various LLMs.

- In addition to theoretical analysis, the authors also use in-distribution data and out-distribution data to do evaluations. The GPT-4 evaluation is adopted to assess the effectiveness of the proposed approach, which is similar to most previous works. The results look good.

**Other Comments Or Suggestions:**

No further comments.

**Other Strengths And Weaknesses:**

Strengths:

- The presentation of this paper is good. It is easy to understand the motivation and the method.

- The results are also good, which can well reflect the effectiveness of the proposed approach.

Weaknesses:

- Despite the good performance, I think the authors should provide more explanations about in what cases the proposed approach fails.

- It would be good, if one of the qualitative examples in the appendix could be moved to the main paper to better see how the proposed approach works compared to the baselines.

**Questions For Authors:**

No further questions.

**Relation To Broader Scientific Literature:**

Match well.

**Theoretical Claims:**

- The authors have provided proofs in the appendix. It seems that there is no problem.

---

> ### Author Rebuttal · Authors · 2025-04-01
>
> We appreciate your positive feedback on our empirical observation, the effectiveness of our approach for various LLMs, and your acknowledgment of both the theoretical analysis and comprehensive evaluations. We will address each of your comments and concerns below and also in our revised manuscript.
>
> ---
>
> > **Q1:** Have the authors attempted to observe whether similar phenomenon happens to larger-sized models?
>
> **RQ1:** We sincerely appreciate your valuable feedback. In response to your suggestion, **we** **extended our experiments to larger models**, specifically Qwen2.5-14B, to further validate our findings. The distribution of energy loss in the final layer of the LLM across various datasets is presented in Figure [R7](https://ibb.co/Fqw8Q2vY). As shown, **the substantial increase in energy loss consistently correlates with reward hacking across all datasets**. This observation **demonstrates the existence of the energy loss phenomenon in larger-scale LLMs** and further solidifies the contribution of our work.
>
> > **Q2:** What would the model performance change when the chain of thoughts strategy is used？
>
> **RQ2:** Thank you for this insightful suggestion. To investigate the impact of the chain of thought (CoT) strategy, we construct a CoT RL dataset, where we append the phrase "Let’s think step by step." to each prompt. The comparison between PPO with CoT and PPO without CoT is shown in Figure [R8](https://ibb.co/21D6CCKd). As demonstrated, **incorporating CoT leads to a significant improvement in RLHF performance.**
>
> We hypothesize that this enhancement is due to the fact that the CoT strategy effectively constrains the output space, guiding the model to generate responses that better align with the user’s input.
>
> > **Q3:** Failure case of the proposed approach.
>
> **RQ3:** Thank you for this comment. Based on our theoretical analysis, the proposed RL regularization term can also be interpreted as a contextual relevance constraint. This means that **our method may not perform as well in cases where the model's responses are required to be highly divergent, but with less emphasis on contextual relevance**. For instance, as shown in the case presented in Figure [R9](https://ibb.co/sp7S2wwx). In the revised version of our paper, we will explore and summarize additional failure cases to provide a more comprehensive and objective analysis.
>
> > **Q4:** It would be helpful to move one of the qualitative examples from the appendix to the main paper to better illustrate how the proposed approach compares to the baselines.
>
> **RQ4:**  Thank you for your suggestion. In the revised version, we will **move one of the qualitative examples from the appendix to the main paper** to better illustrate the advantage of our proposed approach.

---

> > ### Comment · Reviewer_mgdS · 2025-04-09
> >
> > Thanks for the reponses. My concerns have been addressed. In addition, I also read the reviews by other reviewers. It seems that all the reviewers recognize the novelty and the good motivation. Though the explanatory theory does not fully support the argument, this is not a big issue. I agree with Reviewer mmxN about this. So, I lean towards accepting this paper.

---

> > > ### Author Response · Authors · 2025-04-09
> > >
> > > Dear Reviewer mgdS,
> > >
> > > Thank you for your positive feedback and for raising your score. We truly appreciate the time you dedicated to reviewing our paper, as well as your valuable suggestions for our work.
> > >
> > > Best regards,
> > >
> > > The authors of Paper 3248

---

### Official Review · Reviewer_mmxN · 2025-03-14

**Overall Recommendation:** 3

**Summary:**

This work identifies the Energy Loss Phenomenon in RLHF, where increasing energy loss in the final layer of an LLM is linked to reward hacking. To address this issue, the paper proposes EPPO, which penalizes energy loss growth in the RL reward function.

**Claims And Evidence:**

The authors support their claims with both theoretical proofs and experimental validation.

**Essential References Not Discussed:**

N/A.

**Experimental Designs Or Analyses:**

- The paper conducts comprehensive experiments, demonstrating the relationship between energy loss in the final layer and reward hacking, as well as the effectiveness of EPPO in mitigating both.
- In Figure 5, PPO initially performs well but significantly degrades in later RL stages. This suggests that early stopping in PPO training might serve as a potential solution to reward hacking—i.e., halting training at the peak PPO performance. Have the authors compared the performance of an early-stopped PPO model with that of the final EPPO model?

**Methods And Evaluation Criteria:**

- This paper introduces an interesting perspective on the reward hacking problem by uncovering a relationship between the energy loss in the final layer of LLMs and reward hacking. Specifically, the authors connect energy loss with contextual mutual information, which influences reward hacking. I find this to be a compelling research topic.
- While the paper presents experimental evidence (e.g., Figure 4) to support this finding, I find the theoretical explanation somewhat unconvincing. If I understand correctly, Theorem 3 applies to any layer within the LLM, yet the authors emphasize the final layer’s role in reward hacking. Theoretically, wouldn’t this phenomenon also occur in other layers? Additionally, when energy loss surpasses the threshold -$\alpha$, it becomes positively correlated with the upper bound of contextual mutual information, making it difficult to fully reconcile the theoretical framework with the experimental findings. If I have misunderstood any aspects, I would appreciate clarification, as I am open to revising my score.
- EPPO directly incorporates an energy loss growth penalty into the reward function. However, even if a connection between energy loss in the final layer and reward hacking exists, it may only be a symptom rather than the root cause. Reward hacking ultimately stems from flaws in reward design (or, in RLHF, the quality of preference data). If a well-designed reward function prevents PPO-trained LLMs from exhibiting reward hacking, what happens to the energy loss in their final layers? Moreover, what is the fundamental reason behind energy loss growth in PPO training when an ill-structured reward function is used? I believe the authors should further explore these underlying causes.

**Other Comments Or Suggestions:**

Based on the weaknesses and concerns outlined above, I am currently giving this paper a score of 2. However, if the authors can address my concerns and clarify these points, I would be happy to raise my rating.

**Other Strengths And Weaknesses:**

N/A.

**Questions For Authors:**

See the comments above.

**Relation To Broader Scientific Literature:**

In RL research, mitigating reward hacking is a well-studied topic. The authors should broaden their discussion to include approaches such as reward shaping, which can be used to address this issue.

**Theoretical Claims:**

Based on my assessment, there are no obvious flaws in the theoretical proofs.

---

> ### Author Rebuttal · Authors · 2025-03-31
>
> Thank you for recognizing our research perspective. We will address your comments below.
>
> ---
>
> > **Q1:** Does the energy loss phenomenon occur in other layers?
>
> **RQ1:** Thanks for your thoughtful comment. We agree that, in theory,  excessive energy loss at any layer could reduce contextual relevance and contribute to reward hacking. However, **extensive experiments show that only the final layer consistently exhibits a strong trend** across different LLMs and datasets. Evidence is provided in Figures [R1](https://ibb.co/67nmsyF0), [R2](https://ibb.co/50yt1WY), and [R3](https://ibb.co/x8wMFTcS).
>
> We speculate that the final layer, being closest to the output, encapsulates richer semantic information, making it a reliable indicator of overfitting or reward hacking. While earlier layers may show similar trends, their behaviors are influenced by model architecture, training strategies, and data, making patterns less stable and harder to generalize.
>
> > **Q2:** How to reconcile the theoretical framework with the experimental findings?
>
> **RQ2:**  Great question! Our theoretical explanation holds when energy loss stays below the threshold, where **increased loss compresses contextual mutual information, providing a theoretical rationale for the energy loss phenomenon**. Conversely, while increased energy loss exceeding the threshold may raise the upper bound, **it does not necessarily correlate with enhanced contextual relevance.** Why increased energy loss is linked to reward hacking in such scenarios remains an open question.
>
> We would like to clarify that **our main contribution is the empirical findings of the energy loss phenomenon and the corresponding RL regularization design**, while **the theoretical analysis is an exploratory attempt to explain the empirical findings.** In the revised version, we will tone down the theoretical component and highlight the empirical contributions to better reflect the core value of this work.
>
> > **Q3:** The root cause of reward hacking and the role of EPPO in addressing it.
>
> **RQ3:** We agree that reward hacking stems from flaws in reward design. However, in practice—especially in RLHF—it is challenging to obtain a perfect reward model due to overfitting, misspecification, and misgeneralization. In such cases, RL regularization techniques are necessary to guide the model's optimization toward human-desired behavior, even with imperfect reward models. **EPPO is designed to address the limitations of imperfect reward models by incorporating RL regularization.**
>
> > **Q4:** What would happen to the energy loss using a well-designed reward function?
>
> **RQ4:** As discussed in RQ3, obtaining a well-designed (perfect) reward model in RLHF is challenging. To simulate this, we **use a stronger reward model based on LLaMA3-70B as the well-designed function**. The corresponding energy loss and hacking sample distributions after RLHF are shown in Figure [R4](https://ibb.co/7fgH3bc). We observe that **with a well-designed reward model, final-layer energy loss increases only moderately—unlike the sharp rise seen with an ill-structured reward model**.
>
> > **Q5:**  The fundamental reason behind energy loss growth in PPO training when an ill-structured reward function is used.
>
> **RQ5:**  Thanks for your comments. The growth in energy loss is primarily an **empirical observation**, consistently found across various LLMs, datasets, and tasks. To explore its causes, we first provide a theoretical proof that, in certain scenarios, **increased energy loss tends to suppress contextual relevance** (i.e., Theorem 3)—a key aspect of reward hacking. We then empirically demonstrate that **this effect is widely observed across all scenarios** (i.e., Figure 7 in the manuscripts).
>
> We suspect that **the fundamental reason behind energy loss growth lies in overfitting to reward model–favored patterns**—a phenomenon we associate with **reward hacking**. This overfitting prioritizes alignment with the reward model’s biases over accurately capturing user intent, leading to **deterioration in contextual relevance** and a significant reduction in neural activation, manifested as excessive energy loss.
>
> > **Q6:** Comparison with early-stopped PPO.
>
> **RQ6:** Following your suggestion, we **compared EPPO with** **early-stopped PPO** based on GPT-4 evaluations. As shown in Figure [R5](https://ibb.co/67yVdqj3), **EPPO consistently outperforms early-stopped PPO**, demonstrating its stronger resistance to reward hacking.
>
> > **Q7:** Discussion about reward shaping approach in RL research.
>
> **RQ7:** Thanks for your comment. As far as we know, **reward shaping** in RL is typically used to address the **sparse-reward challenge** by providing more informative reward signals, which differs from our focus on **mitigating reward hacking in RLHF** scenarios.
>
> Notably, **EPPO can be seen as a form of reward shaping**, as it uses the internal neural behaviors of the LLM as auxiliary signals to adjust the reward during RL.

---

> > ### Comment · Reviewer_mmxN · 2025-04-09
> >
> > Thanks to the authors for the detailed response.
> > > We suspect that the fundamental reason behind energy loss growth lies in overfitting to reward model–favored patterns—a phenomenon we associate with reward hacking. This overfitting prioritizes alignment with the reward model’s biases over accurately capturing user intent, leading to deterioration in contextual relevance and a significant reduction in neural activation, manifested as excessive energy loss.
> >
> > I believe this insight may be valid. I encourage the authors to expand on this point in their discussion, while substantially trimming the explanatory theory that does not directly support the proposed method. As the authors’ detailed response has addressed several of my concerns, I am willing to raise my score.

---

> > > ### Author Response · Authors · 2025-04-09
> > >
> > > Dear Reviewer mmxN,
> > >
> > > We sincerely thank you for the updated score and your insightful suggestions. We will carefully revise the paper based on your valuable feedback in the new version.
> > >
> > > Best regards,
> > >
> > > The authors of Paper 3248

---

### Decision · Program_Chairs · 2025-05-01

**Decision:**

Accept (poster)

**Comment:**

This paper highlights an interesting phenomenon observed by the authors in RLHF, namely that the energy loss (i.e., the difference in L1 norms of its input and outputs) of the final layer of an LLM increases during training, with large increases being correlated with reward hacking. This leads to a new algorithm (EPPO) penalizing the increase in energy loss, with empirical results showing in particular improved response quality compared to "traditional" RLHF.

All reviewers agree that empirical results are strong, with the main lingering concerns being related to the theoretical analysis that may not be sufficient enough to fully understand this empirical phenomenon and justify the proposed algorithm.

Although I agree it would be great to have even more convincing theoretical results in order to fully understand what's going on, I consider that (1) the empirical results are solid enough to be of interest to other researchers, who may also contribute to investigate this phenomenon in more depth in future work, and (2) the authors did provide some elements of theory that are relevant and at least help build some intuition about their method.

As a result, I consider this paper to be worth presenting at ICML.